# The Role of Place Attachment in the Relationship between Attitudes toward Aging and Subjective Well-Being among Community-Dwelling Older Adults in Taiwan

**DOI:** 10.3390/healthcare12100981

**Published:** 2024-05-09

**Authors:** Jia-Jen Chen, Li-Fan Liu, She-Ming Chen

**Affiliations:** 1Institute of Gerontology, College of Medicine, National Cheng Kung University, Tainan 701, Taiwan; student3841701@gmail.com; 2Department of Architecture, College of Planning and Design, National Cheng Kung University, Tainan 701, Taiwan; chensm@ncku.edu.tw

**Keywords:** age-friendly city/community, attitudes toward aging, mediator, place attachment, subjective well-being, structural equation modeling (SEM)

## Abstract

Subjective well-being presents a societal challenge for vulnerable older adults. This study aims to investigate the mediating role of place attachment in the relationship between attitudes toward aging and subjective well-being among community-dwelling older adults in Taiwan. Two waves of investigations were conducted to examine the interplay between attitudes toward aging, subjective well-being, and place attachment among older adults. In Wave I, 1190 participants were enrolled, revealing predominantly younger cohorts with substantial educational levels. The subsequent Wave II involved 483 participants, maintaining continuity in characteristics. Subjective well-being remained moderate across waves, with prevalent positive attitudes toward aging. Place attachment scores indicated moderate to high associations. After controlling for demographics, structural equation modeling (SEM) in both waves revealed significant positive associations: attitudes toward aging influenced well-being, attitudes toward aging were positively associated with place attachment, and place attachment was positively related to well-being. Mediation testing confirmed the mediating role of place attachment in the relationship between attitudes toward aging and well-being. These findings underscore the important role of place attachment. It is evident that improving attitudes toward aging is an effective intervention which can lead to a better sense of well-being by enhancing place attachment to empower civil society.

## 1. Introduction

Global aging is a prominent phenomenon, particularly evident in Taiwan. Taiwan entered the stage of an aging society in 1993, advanced to an aged society in 2018, and is poised to become a super-aged society by 2025. In 2020, Taiwan’s demographic structure revealed that its proportion of the population aged 15–64 ranked the second highest among countries, behind South Korea. However, projections for 2070 suggest that Taiwan’s proportion in this age group is anticipated to become the second lowest after South Korea. Looking ahead to 2070, demographic forecasts indicate that 4 out of 10 Taiwanese individuals will be elderly, and 1 out of 4 elderly individuals will be over 85 years old [1]. Since the policy of aging in place has arisen around the world, subjective well-being has become a significant social indicator of good health and well-being (UN Sustainable Development Goal 3) as well as of sustainable cities and communities (UN Sustainable Development Goal 11). To improve the lives of older adults and foster healthy aging, the World Health Organization (2021) published the «UN Decade of Healthy Aging 2020–2030» and proposed four goals: (i) change how we think, feel, and act toward age and aging; (ii) ensure that communities foster the abilities of older people; (iii) deliver person-centered, integrated care and primary health services responsive to older people; (iv) provide access to long-term care for older people who need it [2].

At the individual level, theoretical perspectives emphasizing life course development support the possibility of positive attitudes toward aging and positively experiencing the aging process [3]. Extensive research has shown that positive attitudes toward aging might contribute to healthier physical and mental outcomes in older adults [4,5,6,7] through more healthy behaviors [8,9] and attending social and community activities [4]. A systematic review showed that older adults’ perceptions of aging were related to memory and cognitive performance, physical and physiological performance, medical conditions and outcomes, disability, care-seeking, self-rated health, quality of life, and death [10]. In addition, individuals who have fewer negative attitudes toward aging tend to engage in community activities and participate more in their community, and consequently strengthen place attachment [11]. This evidence emphasized attitudes toward aging as a psychological construct accountable for debilitation [12]. Nevertheless, attitudes toward aging become inevitably challenging as a personal resource for successful aging when an individual experiences more chronic diseases or disability.

The well-known “ageing in place” policy posits that older individuals residing in a familiar environment experience a positive influence on their well-being that contributes to favorable experiences in their later years [13]; this is also discussed in the sense of an attachment to place, as a place has certain social connections, security, familiarity, and a sense of identity [14]. The three levels of attachment to place are home, home environment, and the neighborhood. Ageing in place includes not only staying in one’s own home, but also includes remaining in a stable and known environment where people feel they belong. Van Hees et al. (2017) employed a methodology that entails the segmentation of place into socially interconnected facets and physical attributes. The social aspects refer to the place where people live with respect to emotions, memories, experiences, and people, while the physical aspects are related to the function and physical or hard elements of the place [15,16].

In accordance with person–environment theories of aging, individuals residing in environments tailored to their physical, cognitive, and emotional requirements experience elevated levels of well-being [17]. From a community perspective, older adults are socially related to the living environment where they live. Therefore, the emotional bond that individuals form with their surroundings is termed place attachment [18]. This concept intertwines the individual and their environment, facilitating the examination of the “person in place” phenomenon [19]. Attachment to places where people find meaning may contribute more motivation to seek, stay in, protect, and improve their place, and may even lead to more renewal projects [11]. The concept of place attachment has garnered considerable attention, particularly within the realm of planning for aging in place and the cultivation of age-friendly communities [20]. Place attachment is generally a multifaceted concept that characterizes the bonding between people and their particular places, and the definitions of place attachment are inconsistent. For example, place attachment encompasses dimensions such as place identity, place dependence, and social relations within the residential community at a personal level [17]. Aliakbarzadeh et al. (2021) further defined place attachment as a universal feeling among older adults. Nevertheless, previous studies have consistently reported that higher levels of place attachment contribute to better well-being, demonstrating the importance and benefits of place attachment as a psychosocial factor [17,21,22,23,24,25].

Unfortunately, the potential contributions of concerning attitudes toward aging to well-being through place attachment have received little research attention in Chinese culture, including Taiwan. One notable exception was a recent study conducted by Au et al. (2020) [26] which validated the mediating effects of the sense of community among older adults in Hong Kong. Furthermore, empirical evidence regarding the pathway between attitudes toward aging and various psychological dimensions of place attachment (such as place identity, place dependence, and social relations within the community) in relation to subjective well-being is still lacking. To build an age-friendly environment, uncovering the role of attitudes toward aging through place attachment becomes essential to empowering psychosocial community development as a pathway to subjective well-being [27]. Therefore, this study aimed to investigate the association between attitudes toward aging and place attachment in relation to well-being among community-dwelling older adults in Taiwan. Specifically, we used structural equation modeling to explore the relationship between attitudes to aging, place attachment, and subjective well-being, and hypothesized that attitudes toward aging influenced well-being by means of place attachment. Therefore, the hypotheses below were posited based on the previous related literature:

**H1.** 
*Attitudes toward aging have a significant effect on subjective well-being.*


**H2.** 
*Attitudes toward aging are significantly positively associated with place attachment.*


**H3a.** 
*Place attachment has a significant positive association with subjective well-being.*


**H3b.** 
*Place attachment mediates the relationship between attitudes toward aging and subjective well-being.*


## 2. Materials and Methods

### 2.1. Study Sample

The data were collected from the first and second wave of surveys from a university social responsibility program that was a multi-county study including two rural and two urban counties in southern Taiwan. The first wave survey (Wave I) occurred from October 2020 to February 2021, and the second wave survey (Wave II) was conducted from September 2021 to March 2022. In Wave I, we enrolled 1190 community-dwelling older adults aged 50 and above. Subsequently, to assess the various dimensions of place attachment in Wave II, we incorporated a validated place attachment scale [28]. Thus, we recruited 483 individuals for Wave II from the participants of Wave I. Because of the registration law protecting the privacy and security of citizens, a convenience sampling method was used and face-to-face interviews were conducted with the assistance of local community volunteers and interviewers. Before the formal fieldwork, all staff members were required to attend interviewer training. All parties were required to wear face masks during the interviews, which took place with community-dwelling older adults aged 50 years and over. The exclusion criteria were as follows: (a) those who had severe cognitive impairments; (b) those who were bed-ridden; (c) those who were unwilling to disclose their demographics and did not sign an informed consent. Regarding sampling, the rule-of-thumb in the determination of the minimum sample size was to exceed 200 participants, which was considered large [29] when using structural equation modeling. These two cohorts were utilized to evaluate the mediating effect, that is, employing a single-item measure in the Wave I dataset and a formal scale in the Wave II dataset, respectively. The study protocol (A-ER-109-361) was approved by the University Human Research Ethics Committee, and no conflicts of interest were declared.

### 2.2. Measurements

The questionnaire design included aspects regarding lifestyle and health and was simplified to increase the response rate during the COVID-19 pandemic. Measurements used in the study were as follows.

#### 2.2.1. Subjective Well-Being

Subjective well-being was measured by the five-item World Health Organization Well-Being Index (WHO-5), which is a questionnaire that is widely used to assess subjective well-being with a six-point scale (all of the time = 5, most of the time = 4, more than half of the time = 3, less than half of the time = 2, some of the time = 1, and at no time = 0) [30]. A previous study showed that it is a reliable and valid instrument in a community population in Taiwan [31]. The Cronbach’s α for this study was 0.94 in Wave I and 0.92 in Wave II.

#### 2.2.2. Attitudes toward Aging

This study adapted five items of positive attitudes toward aging from the Taiwan Longitudinal Study on Aging using a five-point Likert scale (strongly agree = 5, agree = 4, neither agree nor disagree = 3, disagree = 2, and strongly disagree = 1), where higher scores indicate more positive attitudes to aging. The items included (1) I can still enjoy doing things that interest me; (2) Aging means to me that life is worth living; (3) I feel I can help my family; (4) I feel I am happy and joyful; (5) I feel I am kind and warm. The Cronbach’s α coefficient for the scale was 0.80 in Wave I and 0.78 in Wave II.

#### 2.2.3. Place Attachment

In the Wave I dataset, we used one item of place attachment to define a universal feeling among older adults [20]; thus, we asked the participants to answer “How attached are you to the community where you live?” on a five-point scale (excellent = 5, very good = 4, good = 3, fair = 2, and poor = 1), where higher scores indicate stronger attachment.

The fundamental elements integral to place attachment encompass place definition, place dependence, place bonding, place interaction, and place identity [19]. Consequently, we employed a formal scale to measure these components in the Wave II survey. The place attachment scale [28] comprises three domains with eight items, encompassing (a) Place Identity—(1) How much would you like to live in your current neighborhood? (2) How many good memories do you have in your neighborhood? (3) How many special places in your neighborhood capture your attention? (b) Place Dependence—(4) How relaxed do you feel in this neighborhood? (5) How sad would you feel if you had to leave your neighborhood? (c) Social Relations in the Neighborhood—(6) How friendly are your relationships with your neighbors? (7) To what extent do you assist your neighbors when they are facing difficulties? Responses were recorded on a 5-point Likert scale, ranging from very low to very high [28]. The Cronbach’s α for this scale was 0.8 in the Wave II dataset.

#### 2.2.4. Covariates

The demographic variables were gender, age, educational level, marital status, and the number of chronic diseases. The individuals reported their gender dichotomously (1 = male and 2 = female). Age was measured in years at the end of the month in which they were interviewed. Educational levels were “high school diploma and above” or “below a high school diploma”. Marital status was coded dichotomously (“married” or “other”). Major chronic diseases were self-reported, where those with two or more major chronic diseases were classified as having multimorbidity [32].

### 2.3. Data Analysis

#### Structural Equation Modeling (SEM)

SEM was used to examine the mediation effect linking attitudes toward aging to subjective well-being through place attachment. The measurement model of attitudes toward aging with five indicators and that of subjective well-being with five indicators both showed good model fit. The main paths of interest included the direct effects of attitudes toward aging on subjective well-being and the indirect effects through place attachment. In addition, attitudes toward aging and subjective well-being showed regression on the aforementioned covariates. The mediator (place attachment) also showed regression on covariates. To test the mediation effect, a nonparametric bootstrapping approach was used to obtain estimates for the indirect effect. Nonparametric bootstrapping would resample the raw data and form an empirical distribution of the point estimates to estimate the confidence interval of an indirect effect to assess the significance of the effect [33]. An acceptable SEM had a comparative fit index (CFI) > 0.90, (TLI) > 0.90, a standardized root mean square residual (SRMR) < 0.06, and a root-mean square error of approximation (RMSEA) < 0.08 [34]. All of the analyses were conducted using R-Studio software (2020) (RStudio, Boston, MA, USA, http://www.rstudio.com/), and a maximum likelihood estimation with robust standard errors (MLR) was carried out with the lavaan package.

## 3. Results

### 3.1. Descriptive Statistics

This study was conducted over two distinct waves to collect data on demographic attributes, attitudes toward aging, subjective well-being, and place attachment, as outlined in Table 1. During the initial phase (Wave I), the study enrolled 1190 participants. The gender distribution showed that 43.3% were male and 56.7% were female. Most participants (82.6%) were under the age of 75, indicating a predominance of a relatively younger cohort. Educational achievements were substantial, with 63.3% attaining a high school education or above. Marital status revealed that 72.1% were married or had partners, while 27.9% were either divorced, widowed, or unmarried. Furthermore, 46.3% exhibited multimorbidity, defined as the presence of two or more chronic diseases. In the subsequent survey (Wave II), a targeted subset of 483 participants was recruited, displaying a distribution similar to that observed in Wave I. This continuity in participant characteristics between the two waves enhances the robustness of this study’s dataset and facilitates a comprehensive examination of the dynamics across distinct phases. In the evaluation of subjective well-being, the sample consistently exhibited moderate levels in both Wave I and Wave II. Also, participants reported positive attitudes toward aging in both waves, with mean attitudes indicating a prevailing positive outlook on various aspects related to aging. Turning our attention to the mediator (place attachment), the average place attachment score stood at 3.90 (SD = 0.77) in Wave I, indicating a generally high level of perceived connection to their living environment. In Wave II, including 483 participants, the mean score on the place attachment scale ranged from 3.26 to 4.21, indicating a moderate to high level in this study.

Additionally, Figure 1 depicts the correlations between variables of interest in both Wave I and Wave II, employing the R package “Corrplot” and Pearson’s correlations. Each circle is color-coded in either purple (a positive correlation) or brown (a negative correlation), indicating the sign of the correlation, and the circle size reflects the corresponding correlation coefficients. In Wave I, place attachment showed positive correlations with both subjective well-being and attitudes toward aging. In Wave II, most items on the place attachment scale exhibited positive correlations with subjective well-being and attitudes toward aging, with a few exceptions.

### 3.2. Multilevel Structural Equation Analysis

Figure 2 shows the standardized regression estimates. For Wave I analysis, the model fit the data adequately (χ^2^ (81) = 333.041, *p* = 0.000, CFI = 0.970, TLI = 0.960, SRMR = 0.041, RMSEA = 0.051, and 95% CI of RMSEA [0.045, 0.057]). The standardized factor loading of all items ranged from 0.574 to 0.903. H1 proposes a significant effect of attitudes toward aging on well-being. The model demonstrates a significant effect (β = 0.575, *p* < 0.001); hence, H1 was supported. H2, which hypothesized that attitudes toward aging have a significant positive association with place attachment, was also supported (β = 0.385, *p* < 0.001) in this study. Place attachment was found to be significantly positively related to well-being (β = 0.110, *p* < 0.001); hence, H3a was supported. Additionally, when assessing the impact in the Wave II dataset, the model exhibited acceptable fit to the data (χ^2^ (180) = 492.391, *p* = 0.000, CFI = 0.915, TLI = 0.895, SRMR = 0.073, RMSEA = 0.064, and 95% CI of RMSEA [0.057, 0.071]). Notably, the standardized factor loadings for all items ranged from 0.574 to 0.903, reinforcing the reliability of the measurement model. All estimated pathways were found to be statistically significant, which supported our hypotheses.

Furthermore, H3b involves mediation testing, with place attachment used as the mediator between attitudes toward aging and well-being. A bootstrap mediation analysis was conducted to examine the mediating effect, the results for which can be found in Table 2. If the confidence interval does not include zero, the coefficient would be significantly different from zero. In the Wave I dataset, the estimate of the direct effect of attitudes toward aging on well-being was found to be positive and significant (effect = 0.573, boot SE = 0.028, and 95% bootstrap CI = [0.517, 0.626]). The total effect between attitudes toward aging and well-being was positively significant (effect = 0.615, boot SE = 0.023, and 95% bootstrap CI = [0.566, 0.657]). The indirect mediating effect of attitudes toward aging on well-being was β = 0.042 and was statistically significant (effect = 0.042, boot SE = 0.012, and 95% bootstrap CI = [0.019, 0.066]). In the Wave II dataset, we observed a significant direct effect of attitudes toward aging on well-being (effect = 0.441, boot SE = 0.054, and 95% bootstrap CI = [0.335, 0.547]). The total effect linking attitudes toward aging and well-being remained positively significant (effect = 0.471, boot SE = 0.052, and 95% bootstrap CI = [0.368, 0.570]). There was also a statistically significant indirect mediating effect of attitudes toward aging on well-being, with a β value of 0.029 (effect = 0.029, boot SE = 0.014, and 95% bootstrap CI = [0.003, 0.059]). These findings indicate that some but not all of the effects of attitudes toward aging on well-being are carried through place attachment, hence supporting H3b. It can be inferred from these results that of the 0.615-unit difference in well-being to the 1-unit difference in attitudes toward aging (total effect), 0.042 of that difference showed an effect of attitudes toward aging on place attachment (indirect effect), which in turn influences well-being among older adults. Also, the Wave II data revealed the mediating role (indirect effect = 0.029) of place attachment, exhibiting a pattern consistent with that observed in Wave I (see Table 2).

## 4. Discussion

Well-being enhances one’s ability to age successfully when confronted with the accumulated challenges of older adulthood, as posited by both theoretical and empirical investigations into positive aging [35]. This study surveyed the issue in two waves and the findings in both Wave I and II showed that attitudes toward aging are related to increased subjective well-being among Chinese older adults living in the community, after controlling for gender, age, educational level, marital status, and comorbidities. Since H1 was supported, the main effect of attitudes toward aging on increased subjective well-being was consistent with previous evidence [4,5,6,7], and robust evidence was found which suggested that attitudes toward aging serve as a protective factor for subjective well-being. Based on H2, attitudes toward aging were related to place attachment, which was also consistent with previous findings [22,23,24]. Finally, the hypotheses positing that attitudes toward aging increase the sense of well-being through the mediating effects of place attachment were tested. H3a and H3b were also supported, indicating that attitudes toward aging can enhance place attachment among older adults, and in turn increase subjective well-being. These results provide evidence of the mediating effects of place attachment on the association between attitudes toward aging and subjective well-being.

The association between attitudes toward aging and subjective well-being have been rigorously validated. Previous research showed that positive perceptions of aging prevail among the majority of elderly individuals, while some hold negative views [36]. A positive perception of aging has been recognized as a significant predictor of quality of life in older adults [37]. Our study revealed that attitudes toward aging have a significant impact on the enhancement of well-being, supporting prior research findings [4,5,6,7]. Positive attitudes toward aging emerge as a protective factor for well-being, potentially promoting increased physical activities and social participation [38].

Despite a strong motivation among older adults to maintain positive aging attitudes, concerns related to disease and mortality may contribute to the decline in well-being. It is imperative to recognize the influential role of age, cultural context, personality, and health conditions [36]. Furthermore, positive perceptions of aging encounter significant challenges due to the widespread prevalence of negative stereotypes associated with old age in society [39]. A recent study utilizing the Taiwan Longitudinal Study on Aging (TLSA) dataset revealed that factors detrimental to attitudes toward aging included advanced age, a higher prevalence of comorbidities, living alone, depression, and a lack of independent physical function [40]. Consequently, the difficulty of maintaining positive attitudes toward aging intensifies as individuals grapple with chronic diseases, a heightened risk of disability, or the stereotypes associated with ageism.

Nevertheless, place attachment might provide social resources that help individuals to maintain better physical and mental health. Familiar environments play a constructive role in fostering functional and social competence among older populations [41]. Aging in place for older adults involves an internalized positive feeling [14] that corresponds to remaining in the community and staying independent. It also echoes the dynamic interplay between one’s personal and social self for social identification [42]. Our research findings revealed an association between attitudes toward aging and place attachment, which was consistent with prior studies [22,23,24].

Moreover, we revealed that a heightened sense of place attachment corresponds to increased levels of subjective well-being, reinforcing the significance of place attachment as a robust predictor for the well-being of older adults [17]. A recent study by Han et al. (2021) [43] examined the impact of behavioral, environmental, and intermediate attributes of greenways on the well-being of older adults. Their investigation particularly explored how place attachment could function as an intermediate factor in facilitating aging in place within the Taichung Urban Greenway system. The findings indicated that the quality of greenways, perceived pollution, activities conducted on the greenways, neighborhood social capital, and place attachment significantly influenced the well-being of older adults [43]. Although our study focuses on place attachment within the psychological domain, the importance of environmental attributes in future studies should not be ignored.

Lastly, our findings elucidated that attitudes toward aging not only directly contribute to well-being but also indirectly amplify it through the mechanism of increased place attachment. Regardless of whether a single-item measure [20] or a formal scale [28] was employed to assess place attachment, the mediated effect existed. To our knowledge, there are no studies that explore place attachment as a mediator between positive attitudes toward aging and subjective well-being. Therefore, our explanation regarding this possible pathway is that older adults with optimistic views of aging are likely to promote positive affects [44] and participate more in social and community activities [4]. Additionally, positive social contacts and networks can enhance older adults’ capacity for social adaptation that allows them to access more social resources and support and maintain a sense of belonging. A positive attitude thereby reduces feelings of social isolation or loneliness [45]. In brief, these social ties that provide a sense of belonging, purpose, and/or fulfillment within older adults’ surroundings ultimately influence their well-being [46].

In the quest for optimal well-being, a profound interconnection with the environment unfolds, necessitating a thorough evaluation in alignment with the age-friendly city approach [17]. In accordance with the World Health Organization’s International Technical Meeting on Aging in Place (2022) [41], age-friendly environments are designed to facilitate optimal aging experiences for individuals, allowing them to age well in a location suited to their needs. These environments aim to support ongoing personal development, encourage active contributions to communities, and ultimately promote independence and good health as individuals progress through the aging process [47]. The mediating role of place attachment supported in this study allows an individual to access more social resources in their community, which in turn enhances their sense of well-being. The findings of the present study demonstrated that not only is maintaining positive attitudes toward aging particularly useful for enabling seniors to have higher levels of subjective well-being, but also that interventions that employ age-friendly social resources and increase place attachment would, in turn, enhance subjective well-being among older adults. When aging in place is guided by place attachment, the ensuing sense of belonging plays a pivotal role in mitigating the disruptive effects of life changes associated with old age, including transitions such as retirement and other significant events [48]. Levy (2018) introduced the Positive Education about Aging and Contact Experiences (PEACE) model to enhance the health and well-being of individuals [49], which centers on (a) education about aging, encompassing factual information about the aging process and exposure to positive older role models to dispel negative and inaccurate images of older adulthood; and (b) fostering positive contact experiences with older adults that are individualized, offer or endorse equal status, are cooperative, involve the sharing of personal information, and are sanctioned within the setting. In addition, aging in place affords the opportunity to perceive old age as an integral facet of the life cycle, constituting an ongoing reality augmented by the accumulation of additional years [48]. The predominant healthcare strategy of aging in place accentuates the continual residence of elderly individuals within their community, underscoring a predilection for familiar surroundings as a means to uphold independence and cultivate social support [17]. When an individual engages with a place and feels uneasy about leaving it, an emotional bond is established [20]. Consequently, a pivotal policy consideration when designing age-friendly environments is to not only direct individuals toward positive educational activities but also to establish resources or activities that nurture place attachment. This approach is instrumental in contributing to the overall well-being of older adults.

There are still some limitations in this study. First, due to the household registration act and having to conduct the interviews during the COVID-19 period, only convenience samples could be collected in this study. Future studies may consider surveys in cooperation with local governments, using representative sampling techniques. Secondly, due to our cross-sectional design, future research should warrant more investigation in a causal direction. Finally, the knowledge garnered from this study was constrained by the assessment of place attachment and attitudes toward aging. This limitation stemmed from the inability to delve deeper into various facets, such as place interaction or negative attitudes toward aging, which are pertinent to overall well-being. Hence, future studies should encompass more comprehensive inquiries to discern and compare various facets, thereby further addressing these aspects.

## 5. Conclusions

This study contributes to the field of research on positive aging by investigating the intricate relationship between attitudes toward aging, place attachment, and subjective well-being among Chinese older adults in community settings. Notably, place attachment emerges as a crucial factor influencing the well-being of older adults in our two waves of analysis. The positive correlation between attitudes toward aging and place attachment underscores the latter’s mediating role in connecting attitudes toward aging to subjective well-being, emphasizing the significance of establishing age-friendly environments. Interventions aimed at reinforcing positive attitudes toward aging should prioritize the cultivation of place attachment and built-in educational activities. Fostering a sense of belonging and emotional connection to the community equips individuals with the capability to navigate the challenges of an ageist society, in turn making a substantial contribution to their overall well-being.

## Figures and Tables

**Figure 1 healthcare-12-00981-f001:**
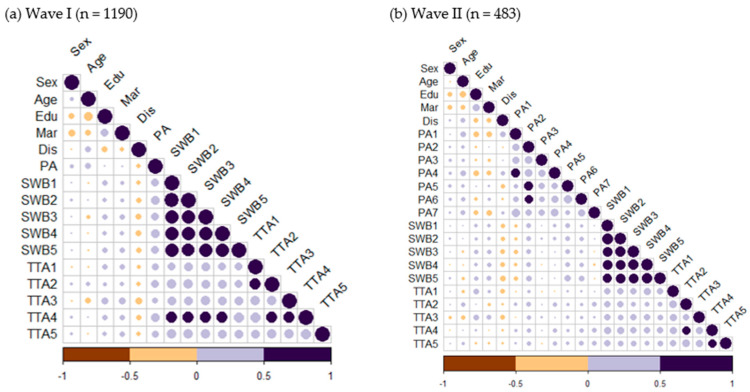
Correlation of subjective well-being with all variables in Wave I and Wave II dataset. Note: for clarity, abbreviations were employed; sex (1: male; 2: female), age (1: <75 years old; 2: 75 and above), education (1: <high school education; 2: high school and above), marital status (MAR; 1: others and 2: married), multimorbidity (DIS; 1: no and 2: yes), place attachment (PA indicates a single item; formal scales consisted of PA1, PA2, PA3, PA4, PA5, PA6, and PA7), variables related to subjective well-being (SWB1, SWB2, SWB3, SWB4, and SWB5), and attitudes toward aging (TTA1, TTA2, TTA3, TTA4, and TTA5).

**Figure 2 healthcare-12-00981-f002:**
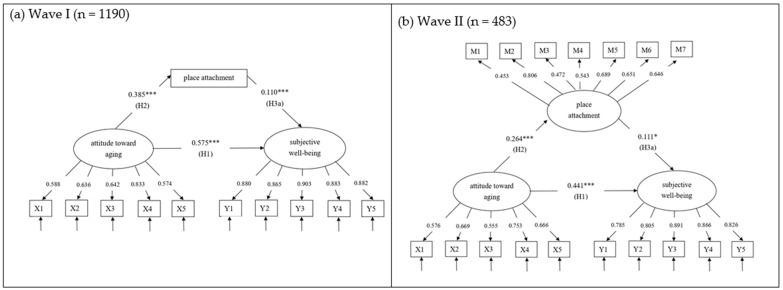
Standardized SEM estimates of attitudes toward aging, place attachment, and subjective well-being. * *p* < 0.05; *** *p* < 0.001.

**Table 1 healthcare-12-00981-t001:** Basic characteristics of the participants.

	Wave I (n = 1190)		Wave II (n = 483)	
Variable	n/Mean	%/SD	n/Mean	%/SD
Gender				
Male	515	43.3	207	42.9
Female	675	56.7	276	57.1
Age (range: 50–99)				
Less than 75 years old	983	82.6	404	83.6
75 years old or over	207	17.4	79	16.4
Education level				
Below high school education	437	36.7	123	25.5
High school or above	753	63.3	360	74.5
Marital status				
Divorced, widowed, or unmarried	332	27.9	127	26.3
Married/has partners	858	72.1	356	73.7
Number of chronic diseases				
Less than two	639	53.7	314	65.0
Two or more	551	46.3	169	35.0
Subjective well-being (range: 0–5)				
I feel cheerful and in good spirits	3.54	1.18	3.78	0.98
I feel calm and relaxed	3.51	1.15	3.81	1.00
I feel active and vital	3.33	1.29	3.63	1.16
I wake up feeling fresh and rested	3.37	1.26	3.65	1.15
My daily life is filled with things that interest me	3.17	1.38	3.52	1.25
Attitudes toward aging (range: 1–5)				
I can still enjoy doing things that interest me	3.67	0.76	3.69	0.70
Aging means to me that life is worth living	3.77	0.79	3.86	0.69
I feel I can help my family	3.65	0.89	3.78	0.74
I feel I am happy and joyful	3.72	0.81	3.88	0.72
I feel I am kind and warm	3.75	0.74	3.87	0.68
Wave I: single item of place attachment (range: 1–5)	3.91	0.77	-	-
Wave II: formal scale of place attachment (range: 1–5)				
How much would you like to live in your current neighborhood?	-	-	3.86	0.68
How many good memories do you have in your neighborhood?	-	-	3.86	0.72
How many special places in your neighborhood capture your attention?	-	-	3.26	0.79
How relaxed do you feel in this neighborhood?	-	-	4.08	0.59
How sad would you feel if you had to leave your neighborhood?	-	-	3.88	0.68
How friendly are your relationships with your neighbors?	-	-	4.21	0.65
To what extent do you assist your neighbors when they are facing difficulties?	-	-	4.07	0.58

Note: the symbol “-” is used to denote situations where the information is not applicable.

**Table 2 healthcare-12-00981-t002:** The bootstrap point estimates for all of the effects of the mediation model between attitudes toward aging and subjective well-being through place attachment.

Effects	Point Estimates (β)	Standard Error (SE)	95% CIs of Point Estimates	
			Lower Bound	Upper Bound
Wave I (n = 1190)				
direct effect	0.573	0.028	0.517	0.626
indirect effect	0.042	0.012	0.019	0.066
total effect	0.615	0.023	0.568	0.657
Wave II (n = 483)				
direct effect	0.441	0.054	0.335	0.547
indirect effect	0.029	0.014	0.003	0.059
total effect	0.471	0.052	0.368	0.570

Note: unstandardized regression coefficients (β), standard error (SE), and 95% confidence intervals (CIs, lower and upper bounds) in 5000 bootstrap samples.

## Data Availability

The data that support the findings of this study are available from the corresponding author upon reasonable request.

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
