# Peer review of "The Role of Place Attachment in the Relationship between Attitudes toward Aging and Subjective Well-Being among Community-Dwelling Older Adults in Taiwan"

_healthcare, 2024, doi:10.3390/healthcare12100981_

Round 1

Reviewer 1 Report

Comments and Suggestions for Authors

It is a good article. The topic of the article is interesting.

The introduction does not have a complete path to the gap of knowledge. It is necessary for the authors to state more precisely what theories and studies are conceivable based on the relationship between the variables.

The discussion lacks reasons about the why of the results, and it is better for the discussion to have content about Taiwan's social conditions.

Comments on the Quality of English Language

In the text, there are some errors in grammar and word choice.

e.g. 

At the individual level, theoretical perspectives emphasizing life course development support the possibility of positive attitudes toward aging and positively experiencing the aging process [3]. Extensive research had shown that positive attitudes toward aging might contribute to healthier physical and mental outcomes in older adults [4-7], engage in more healthy behaviors [8,9], and social and community activities [4]. A systematic review showed that older adults’ perceptions of ageing were related to memory and cognitive performance, physical and physiological performance, medical conditions and outcomes, disability, care-seeking, self-rated health, quality of life and death [10]. In addition, individuals who endorsed less negative attitude of aging tend to engage in community activities and spend more participation in their community, and consequently strengthen 55the place attachment [11]. These evidences emphasized attitudes toward aging as psychological construct was accountable for debilitation [12]. 

Author Response

Response: Thank you for your positive feedback and insightful comments on our manuscript. We appreciate your thorough review and have taken your suggestions into consideration to improve the quality of our work. We are willing to send for English editing if necessary. At the moment, we have carefully considered your comments and made the following revisions to the manuscript:

Introduction Section: We have revised the statement in the Introduction section to make it smoother. Please see Lines 72-91 in the revised manuscript. The revised paragraphs are as follows:

In accordance with person-environment theories of aging, individuals residing in environments tailored to their physical, cognitive, and emotional requirements experience elevated levels of well-being [22]. From a community perspective, older adults are socially related to the living environment where they live. Therefore, the emotional bond individuals form with their surroundings is termed place attachment [17]. This concept intertwines the individual and their environment, facilitating the examination of the “person in place” phenomenon [18]. Attachment to places when people who find meaning in the place were attached may contribute more motivation to seek, stay in, protect, and improve their place, and even more renewal projects [11]. The concept of place attachment has garnered considerable attention, particularly within the realm of planning for aging in place and the cultivation of age-friendly communities [19]. A relevant research in Netherlands identified the positive relationship between self-stereotyping and self-anchoring in society [20]. Place attachment is generally a multifaceted concept that characterizes the bonding between people and their particular places. There are inconsistent definitions of place attachment. For example, place attachment encompasses dimensions such as place identity, place dependence, and social relations within the residential community at a personal level [22]. Aliakbarzadeh et al. (2021) further defined place attachment as a universal feeling among older adults. Nevertheless, previous studies consistently reported that individuals who had higher levels of place attachment contribute better well-being demonstrating the importance and benefits of place attachment as psychosocial factor [21-26].

Discussion Section: We have added a new statement and statement about Taiwan’s study to make it consistent with the previous perspective. Please see Lines 317-320 & Lines 342-355 in the revised manuscript. The revised paragraph is as follows:

Despite a strong motivation among older adults to maintain positive aging attitudes, concerns related to disease and mortality may contribute to the decline of the well-being. It is imperative to recognize the influential role of age, cultural context, personality, and health conditions [35]. Furthermore, positive perceptions of aging encounter significant challenges due to the widespread prevalence of negative stereotypes associated with old age in society [38]. A recent study utilizing the Taiwan Longitudinal Study on Aging (TLSA) dataset revealed that factors detrimental to attitudes toward aging included advanced age, a higher prevalence of co-morbidities, living alone, depression, and dependence on physical function [Lee et al., 2022]. Consequently, the difficulty of maintaining positive attitudes toward aging intensifies as individuals grapple with chronic diseases, heightened risk of disability, or the stereotypes associated with ageism.

Lastly, our findings elucidated attitudes towards aging not only directly contributed to well-being but also indirectly amplified it through the mechanism of increased place attachment. Regardless of whether a single-item measure [19] or a formal scale [28] was employed to assess place attachment, the mediated effect existed. To our knowledge, there are no studies to explore place attachment as a mediator between positive attitudes toward aging and subjective well-being. Therefore, our explanation regarding this possible pathway is that older adults with optimistic views of aging are likely to promote positive affect [Menkin et al, 2017] and participate more in social and community activities [4]. Additionally, positive social contacts and networks can enhance older adults’ capacity for social adaptation that access more social resources and support and allow them to maintain a sense of belonging. Positive attitude thereby reduces feelings of social isolation or loneliness [Sun et al, 2024]. In brief, these social ties that provide a sense of belonging, purpose, and/or fulfillment within older adults’ surroundings ultimately influence their well-being [Fuller-Iglesias, 2015].

New References:

  • Lee SH, Yeh CJ, Yang CY, Wang CY, Lee MC. Factors Associated with Attitudes toward Aging among Taiwanese Middle-Aged and Older Adults: Based on Population-Representative National Data. Int J Environ Res Public Health. 2022, 24;19(5),2654.
  • Sun S, Wang Y, Wang L, Lu J, Li H, Zhu J, Qian S, Zhu L, Xu H. Social anxiety and loneliness among older adults: a moderated mediation model. BMC Public Health. 2024, 15;24(1), 483.
  • Fuller-Iglesias HR. Social ties and psychological well-being in late life: the mediating role of relationship satisfaction. Aging Ment Health. 2015, 19(12), 1103-1112.

Grammar and Language Errors: We have fixed the grammar errors in the highlighted paragraphs. Please see Lines 46-59 in the revised manuscript. The revised paragraphs are as follows:

At the individual level, theoretical perspectives emphasizing life course development support the possibility of positive attitudes toward aging and positively experiencing the aging process [3]. Extensive research has shown that positive attitudes toward aging might contribute to healthier physical and mental outcomes in older adults [4-7], through more healthy behaviors [8,9], and attending social and community activities [4]. A systematic review showed that older adults’ perceptions of aging were related to memory and cognitive performance, physical and physiological performance, medical conditions and outcomes, disability, care-seeking, self-rated health, quality of life, and death [10]. In addition, individuals who have less negative attitude toward aging tend to engage in community activities with more participation in their community, and consequently strengthen the place attachment [11]. This evidence emphasized attitudes toward aging as a psychological construct were accountable for debilitation [12]. Nevertheless, attitudes toward aging become inevitably challenging as a personal resource for successful aging when an individual experiences more chronic diseases or disability.

Thank you once again for your valuable input to enhance the quality of our work!

Reviewer 2 Report

Comments and Suggestions for Authors

I am of two minds with this article.  This is a proficient article that uses appropriate and common statistical modeling and graphics.  The author's hypothesis were predictable and based in a sound literature review.  Yet, the article did not offer any new insights, theoretical perspectives or thoughtful discussion.  They found what everyone else has found.  Confirmation of previous research has value but is often generates little reader interest.  In the end the authors conclude that positive attitudes toward aging and positive attachment to a location generates well being.  I believe this has been well known for many decades.   

Author Response

Response: Thank you for your comments on our manuscript. We have carefully considered your comments and made the following revisions to the manuscript:

Discussion Section: We have added a new statement about Taiwan’s study to make it consistent with the previous perspective. Please see Lines 342-355 in the revised manuscript. The revised paragraph is as follows:

Lastly, our findings elucidated attitudes towards aging not only directly contributed to well-being but also indirectly amplified it through the mechanism of increased place attachment. Regardless of whether a single-item measure [19] or a formal scale [28] was employed to assess place attachment, the mediated effect existed. To our knowledge, there are no studies to explore place attachment as a mediator between positive attitudes toward aging and subjective well-being. Therefore, our explanation regarding this possible pathway is that older adults with optimistic views of aging are likely to promote positive affect [Menkin et al, 2017] and participate more in social and community activities [4]. Additionally, positive social contacts and networks can enhance older adults’ capacity for social adaptation that access more social resources and support and allow them to maintain a sense of belonging and positive attitude, thereby reducing feelings of social isolation or loneliness [Sun et al, 2024]. In brief, these social ties that provide a sense of belonging, purpose, and/or fulfillment within older adults’ surroundings ultimately influence their well-being [Fuller-Iglesias, 2015].

New References:

  • Lee SH, Yeh CJ, Yang CY, Wang CY, Lee MC. Factors Associated with Attitudes toward Aging among Taiwanese Middle-Aged and Older Adults: Based on Population-Representative National Data. Int J Environ Res Public Health. 2022, 24;19(5),2654.
  • Sun S, Wang Y, Wang L, Lu J, Li H, Zhu J, Qian S, Zhu L, Xu H. Social anxiety and loneliness among older adults: a moderated mediation model. BMC Public Health. 2024, 15;24(1), 483.
  • Fuller-Iglesias HR. Social ties and psychological well-being in late life: the mediating role of relationship satisfaction. Aging Ment Health. 2015, 19(12), 1103-1112.

Thank you once again for your constructive feedback!

Reviewer 3 Report

Comments and Suggestions for Authors

Title:

It is suggested to change "psychological well-being" to "subjective well-being", since it is the term used throughout the article.

Introduction:

It is well prepared and understandable. Provides a background review to contextualize the study, underscoring its importance. Furthermore, it clearly defines the objective of the study and presents three hypotheses.

Materials and Methods: 

It adequately describes the aspects necessary to corroborate the methodological rigor of the study: population, variables, instruments and statistical analyzes carried out.

Results:

The results are presented coherently and following a logical sequence, with the inclusion of two tables and two figures that support the hypotheses proposed. However, it is suggested to improve the description of figure 2:

-On page 5, line 216, it is mentioned: "Each circle was color-coded in either purple or brown, indicating the sign of the correlation." It would be helpful to specify which color represents a positive correlation and which a negative correlation.

-Additionally, between lines 217 and 220 on page 5, it would be beneficial to specify that SWB1, SWB2, SWB3, SWB4, SWB5, TTA1, TTA2, TTA3, TTA4, and TTA5 are the ranges of variables related to subjective well-being and attitude towards aging. This clarification can be added either in the text or as a footnote to the figure to enhance reader comprehension.

Discussion:

The discussion addresses the results obtained in relation to the hypotheses proposed, comparing them with previous studies. Additionally, the study's limitations are highlighted, and areas for future research are proposed.

Author Response

Response: Thank you for your detailed review and valuable suggestions. We have carefully considered your comments and made the following revisions to the manuscript:

  • Title: We have changed the term "psychological well-being" to "subjective well-being" throughout the article, as suggested.
  • Introduction: Thank you for your comments
  • Materials and Methods: Thank you
  • Results:
  • The description of Figure 2 has been improved as follows: We have included the statement “Each circle was color-coded either in purple (indicating a positive correlation) or brown (indicating a negative correlation)” (see Lines 224-225 in the revised manuscript).
  • Additionally, we have added the clarification “place attachment (PA indicates a single item; formal scales consisted of PA1, PA2, PA3, PA4, PA5, PA6, and PA7), subjective well-being (SWB1, SWB2, SWB3, SWB4, and SWB5) encompasses variables related to subjective well-being, while attitude toward aging (TTA1, TTA2, TTA3, TTA4, and TTA5) represents attitudes toward aging” in the footnote of Figure 1 (see Figure 1 in the revised manuscript).
  • Discussion: Thank you for your comments

We believe these changes have enhanced the clarity and comprehensibility of the manuscript. Thank you once again for your constructive feedback!
